# MaskGT: Learning Task-Adaptive Connectivity in Graph Transformers

## Abstract

Graph Transformers (GTs) enable all-to-all interactions, but the optimal connectivity is task-dependent: some problems favor sparse, topology-aligned message passing, while others need global attention. We propose MASKGT, a GT-agnostic module that learns a *discrete sparse gate* over attention edges. By learning which node pairs may communicate *within* self-attention, MASKGT injects a task-adaptive relational inductive bias without fully committing to the input adjacency. Across synthetic and real-world benchmarks, MASKGT improves performance and robustness by suppressing spurious interactions under structural noise, and enables parameter-efficient multi-task and transfer by localizing task-specific structure in the mask while reusing a shared backbone. These results position MASKGT as a step toward more general-purpose graph models.

## 1 Introduction

Since their inception, message-passing graph neural networks (GNNs) (Gori et al., 2005; Scarselli et al., 2008; Kipf & Welling, 2017; Hamilton et al., 2017; Velickovic et al., 2018) have been the default architecture for learning on graph-structured data. The central mechanism in GNN is *locality*: node representations are computed by repeatedly aggregating messages from adjacent nodes. While effective, this constrained interaction pattern also leads to well-known limitations, including difficulty capturing long-range dependencies, over-smoothing (Li et al., 2018; NT & Maehara, 2019), over-squashing (Alon & Yahav, 2021; Di Giovanni et al., 2023), and limited expressivity under graph isomorphism tests such as the Weisfeiler–Lehman framework (Xu et al., 2019; Morris et al., 2019).

Graph Transformers (GTs) (Ying et al., 2021; Zhang et al., 2020; Rampášek et al., 2022) address these issues by replacing neighborhood aggregation with self-attention, enabling node-to-node interactions beyond the observed graph topology. With suitable structural and positional encodings, GTs can in principle be more expressive than message-passing GNNs (Kreuzer et al., 2021; Deng et al., 2024). Coupled with the success of transformers in language (Achiam et al., 2023) and vision (Dosovitskiy et al., 2021), this has motivated a growing body of work adapting transformer architectures to graphs (He et al., 2023).

Yet, despite this theoretical appeal, GTs have not consistently replaced GNNs in practice. Classical GNNs can outperform GTs on many standard node classification benchmarks (Luo et al., 2024), and restricting attention to the input topology can outperform dense, all-to-all attention in several settings (Dwivedi & Bresson, 2021). Many strong GT variants therefore reintroduce sparsity and graph bias in different forms: some prepend message-passing layers (Ying et al., 2021; Mialon et al., 2021; Wu et al., 2023; Tang et al., 2025), others combine local message passing with global attention within each layer (Rampášek et al., 2022), and several methods treat attention sparsity as a tunable design choice via hyperparameters or architectural constraints (Kreuzer et al., 2021; Dwivedi & Bresson, 2021; Deng et al., 2024). These results suggest that sparse message passing *relational inductive bias* (Battaglia et al., 2018) remains a key ingredient even in transformer-style models.

At the same time, the *right* relational bias is not universal. The optimal trade-off between topology-aligned sparsity (GNN-like) and global attention (GT-like) varies across tasks and datasets: for example, PTC_MR

and MUTAG benefit more from complete attention, whereas OGBG-MOLHIV and ZINC favor sparse attention, despite all being molecular graphs (cf. SPARSE vs. NO-MASK in Fig. 3 and Fig. 4a; also see Appendix A.1). Moreover, even when sparsity helps, the best interaction pattern need not coincide with the observed adjacency: graphs may be noisy or incomplete, and edge semantics can shift across tasks (e.g., when an equally valid graph is derived from the complement). In these regimes, committing to the input topology can mislead, while always using dense attention can also be sub-optimal.

These observations motivate a simple question: *can we learn modular, task-specific attention connectivity?* We propose MASKGT, a simple, GT-agnostic mechanism that adds a discrete gate over attention edges (Fig. 1). MASKGT learns which node pairs may exchange information *within* self-attention, making the attention wiring a learnable, task-dependent object. Unlike sparse-attention or topology-restricted GTs that predefine the interaction set, and unlike graph structure learning (GSL) methods that output a separate graph for a downstream predictor, MASKGT adapts connectivity *inside* the transformer. Conceptually, it factorizes a GT layer into a shared representation backbone and a lightweight mask module that selects a task-appropriate communication pattern.

MASKGT is inspired by masking-based parameter-efficient adaptation (Mallya et al., 2018; Wortsman et al., 2020; Xue et al., 2022), but it masks *connectivity* rather than weights: it learns which node pairs may communicate, directly shaping relational inductive bias at the level of information flow. This modularity is especially useful for graphs, where edge semantics vary widely across domains and tasks and a single sparsity pattern is unlikely to fit all settings. By localizing task-specific structure in a lightweight masking module, MASKGT enables a shared GT backbone to be reused while adapting interaction structure, supporting multi-task learning and transfer and moving toward more general-purpose graph models.

**Contributions.**

- **MaskGT: task-adaptive connectivity in attention.** We introduce a GT-agnostic discrete mask over attention edges that interpolates between topology-biased sparsity and global attention. We include our code as supplementary materials, and will be made public upon acceptance.

- **Evidence under structure mismatch.** On synthetic denoising and real benchmarks, MASKGT suppresses spurious edges and improves accuracy/robustness over strong GT baselines.

- **Modular adaptation.** Task-specific structure is captured by the mask module, enabling a shared GT backbone for multi-task and transfer learning.

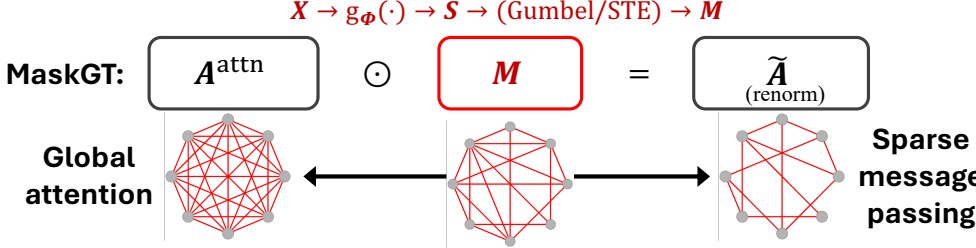

Figure 1: MaskGT learns a discrete mask $\mathbf{M}$ that gates which node pairs may exchange information inside a graph transformer, producing task-adaptive sparse connectivity from an otherwise global attention pattern.

## 2 Preliminaries

We consider a graph $\mathcal{G} = (\mathcal{V}, \mathcal{E})$ with $|\mathcal{V}| = n$ nodes and node features $\mathbf{h}_i \in \mathbb{R}^d$. We denote the (input) neighborhood by $\mathcal{N}(i) = \{j : (i, j) \in \mathcal{E}\}$.

**Connectivity as attention support.** For transformer-style updates, it is useful to distinguish *which* node pairs are allowed to interact from *how strongly* they interact. We therefore define a *candidate interaction set* $\mathcal{C}(i) \subseteq \mathcal{V}$. Standard dense attention uses $\mathcal{C}(i) = \mathcal{V}$, while topology-restricted variants use $\mathcal{C}(i) = \mathcal{N}(i)$.

## 2.1 Message-passing Graph Neural Networks (GNNs)

A message-passing GNN updates node representations by aggregating information from $\mathcal{N}(i)$:

$$\mathbf{h}_i^{(l)} = \text{UPDATE}^{(l)}\Big(\mathbf{h}_i^{(l-1)}, \ \text{AGGREGATE}^{(l)}(\{\text{MESSAGE}^{(l)}(\mathbf{h}_i^{(l-1)}, \mathbf{h}_j^{(l-1)}) : j \in \mathcal{N}(i)\})\Big). \tag{1}$$

Various GNNs, such as GCN (Kipf & Welling, 2017), GraphSAGE (Hamilton et al., 2017) and GIN (Xu et al., 2019), can be described under this framework by defining different message, aggregate and update functions. For graph-level tasks, a permutation-invariant readout produces $\mathbf{h}_{\mathcal{G}} = \text{READOUT}(\{\mathbf{h}_i^{(L)}\}_{i \in \mathcal{V}})$.

## 2.2 Graph Transformers (GTs)

GTs replace neighborhood aggregation with self-attention over a candidate set $\mathcal{C}(i)$. Standard transformer uses dense attention: $\mathcal{C}(i) = \mathcal{V}$. With $H$ heads and per-head dimension $d_k$, the attention weights are

$$a_{ij}^{k,l} = \text{softmax}_{j \in \mathcal{C}(i)}\left(\frac{(\mathbf{W}_Q^{k,l}\mathbf{h}_i^{(l-1)})^\top(\mathbf{W}_K^{k,l}\mathbf{h}_j^{(l-1)})}{\sqrt{d_k}}\right), \tag{2}$$

and node $i$ is updated by aggregating values from $j \in \mathcal{C}(i)$:

$$\mathbf{h}_i^{(l)} = \mathbf{W}_O^l \mathop{\Big\|}_{k=1}^{H} \Big(\sum_{j \in \mathcal{C}(i)} a_{ij}^{k,l} \ \mathbf{W}_V^{k,l}\mathbf{h}_j^{(l-1)}\Big). \tag{3}$$

Positional/structural encodings may be added to inject topology information, such as Laplacian positional encodings (Dwivedi & Bresson, 2021; Kreuzer et al., 2021), which use eigenvectors of the graph Laplacian, and random-walk positional encodings (Li et al., 2020), which use statistics of random-walk transition probabilities. We denote them by $\boldsymbol{\lambda}_i$ and write $\tilde{\mathbf{h}}_i^{(0)} = \mathbf{h}_i^{(0)} + \boldsymbol{\lambda}_i$.

## 2.3 Connectivity interpolates between GNNs and GTs

GNNs and GTs differ primarily in the *support* of interactions (Appendix A.2). If $\mathcal{C}(i) = \mathcal{N}(i)$, transformer updates reduce to topology-based message passing (e.g. sparse attention). If $\mathcal{C}(i) = \mathcal{V}$, they recover dense self-attention on a fully connected graph. Our method will leverage this view by learning a task-adaptive support $\mathcal{C}(i)$ (equivalently, a mask) inside attention.

# 3 MaskGT

We introduce MASKGT, a GT-agnostic module that learns *which node pairs may interact* inside self-attention, thereby injecting a task-adaptive relational inductive bias without hard-coding sparsity. Building on the view in Sec. 2.2 that a GT is defined by both (i) attention *weights* and (ii) attention *support* $\mathcal{C}(i)$, MASKGT augments a standard GT with a *mask generator* that predicts a binary support matrix $\mathbf{M} \in \{0,1\}^{n \times n}$, where $\mathbf{M}_{ij} = 1$ indicates that node $i$ is permitted to attend to node $j$. The mask therefore prunes the attention graph: messages can only flow along edges selected by $\mathbf{M}$, and are still weighted by the usual attention coefficients. In this way, MASKGT learns task-specific connectivity that can interpolate between topology-biased sparse interactions and global attention, while keeping the underlying GT backbone unchanged. The mask can be shared or vary across heads and layers (e.g., $\{\mathbf{M}^{k,l}\}_{k=1,l=1}^{H,L}$). For clarity, we present the single-mask case; the per-head and/or per-layer variants follow directly by repetition.

**Mask generator.** The mask generator is a learnable module $g_{\boldsymbol{\Phi}}$ with parameters $\boldsymbol{\Phi}$ that maps node representations to pairwise logits over candidate interactions. Let $\mathbf{H} \in \mathbb{R}^{n \times d}$ denote the node features input to the generator. It outputs a score matrix $\mathbf{S} = g_{\boldsymbol{\Phi}}(\mathbf{H}) \in \mathbb{R}^{n \times n}$, where $\mathbf{S}_{ij}$ reflects the preference to retain information flow (i.e., message passing) between nodes $i$ and $j$. The pairwise score $\mathbf{S}_{ij}$ can be determined in multiple ways, for example via: (i) dot-product similarity after a learned projection, (ii) an MLP applied to pairwise features, or (iii) an attention-based scoring module (as in meta-attention for ViT (Xue et al., 2022)). In our experiments, we instantiate $g_{\boldsymbol{\Phi}}$ as a 2-layer MLP that takes concatenated node features as input (Appendix B).

**Masking as structural regularization.** Masking is complementary to attention weighting: attention controls *how strongly* nodes communicate, while the mask controls *which* node pairs may communicate. In dense attention, all node pairs remain possible communication channels even if some receive small weights, so the model still searches over a dense relational hypothesis space. By learning a binary support $\mathbf{M}$, MASKGT restricts information flow to a learned sparse communication graph, thereby limiting which pairwise interactions can influence the model's predictions through attention. Thus, masking acts as structural regularization by constraining the relational hypothesis space, $\mathcal{H}_{\mathrm{mask}} \subset \mathcal{H}_{\mathrm{dense}}$.

**Binary mask.** We obtain a discrete mask $\mathbf{M} \in \{0,1\}^{n \times n}$ from $\mathbf{S}$. To enable gradient-based learning with discrete decisions, we use a Gumbel–Softmax (Concrete) relaxation (Maddison et al., 2017) together with a straight-through estimator (STE) (see Appendix B.1), resulting in a binary mask in the forward pass and a differentiable surrogate in the backward pass. The mask is then applied to prune attention connectivity:

$$\tilde{\mathbf{A}}^{k,l} = \mathrm{normalize}(\mathbf{M} \odot \mathbf{A}^{k,l}), \tag{4}$$

where $\mathbf{A}^{k,l} \in \mathbb{R}^{n \times n}$ denotes the post-softmax attention weights from Eq. 2 and $\odot$ is element-wise multiplication (Fig. 1). The normalization is to account for the masked edges.

**Optimization.** We train $g_{\mathbf{\Phi}}$ either (i) *jointly* with the GT backbone end-to-end (Sec. 5.1), or (ii) *post hoc* by freezing a pretrained backbone and optimizing only the mask generator (and task head) for adaptation (such as multi-task (Sec. 5.3) and transfer learning (Sec. 5.4)). In all cases, $\mathbf{\Phi}$ is learned by backpropagating the task loss through the masked attention computation.

## 4 Related Work

**Graph structure learning (GSL).** GSL improves prediction of GNN by modifying or learning graph connectivity. Existing methods broadly fall into similarity-/attention-based approaches that reweight/add/remove edges using learned similarities (Velickovic et al., 2020; Chen et al., 2020; Huang et al., 2020; Wu et al., 2022; Gu et al., 2023; In et al., 2024; Zhao et al., 2023) and direct approaches that optimize adjacency variables with regularization or priors (Zhang et al., 2019; Franceschi et al., 2019; Hasanzadeh et al., 2020; Jin et al., 2020; Wang et al., 2021). MASKGT is closest to the former, but rather than learning a new graph for a downstream GNN, it learns a task-specific sparse connectivity inside the transformer by gating attention edges, directly adapting the GT interaction pattern. Unlike most GSL methods that learn structure only through joint training with the predictor, MASKGT also supports *post hoc* adaptation by training only the mask (and head) while keeping the pretrained GT fixed (Sec. 5.4).

**Parameter masking.** Masking is often used for compression/efficiency, closely related to pruning (Le-Cun et al., 1989; Han et al., 2016). More related are masking methods for parameter-efficient adaptation: Piggyback (Mallya et al., 2018) and supermasks (Wortsman et al., 2020) learn binary masks that select task-specific subnetworks while keeping backbone weights fixed. Closest to our work, meta-attention (Xue et al., 2022) masks attention connections for transfer in vision transformers. MASKGT extends this idea to graphs by masking *node-pair interactions* in GT self-attention, enabling task-adaptive relational bias while keeping the GT backbone shared.

## 5 Experiments

We empirically evaluate whether MASKGT (i) learns a task-appropriate relational inductive bias by selecting task-relevant connectivity—validated through controlled denoising on synthetic graphs and performance on real-world benchmarks where edge semantics are unknown—and (ii) improves adaptivity in multi-task and transfer settings where edge semantics differ across tasks or domains. Beyond the synthetic graph dataset described here, details of the other datasets are provided in Appendix C.

**Model and training protocol.** Because MASKGT is agnostic to the choice of backbone GT, we adopt the classical GT from Dwivedi & Bresson (2021) as our base model across all experiments to minimize confounding effects from architectural variants. The mask generator used is a lightweight 2-layer MLP

with a ReLU nonlinearity, trained jointly with the backbone. We learn a discrete edge-level gate via the Gumbel-Softmax estimator with temperature $\tau = 0.67$. We also augment the node features with random walk positional encoding across all our experiments. Across all experiments, the most direct ablation is no mask (the same GT without masking). We report mean and standard deviation over 10 runs with different random seeds. Full hyperparameters and additional implementation details are provided in Appendix D.

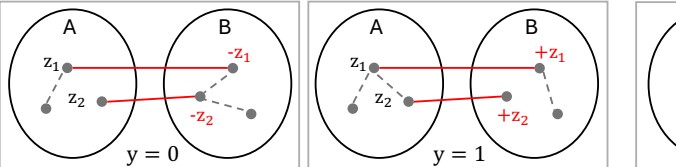 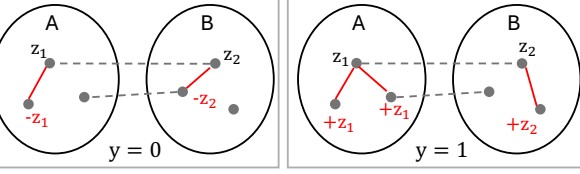

(a) Heterophilic (cross-partition) agreement signal.  (b) Homophilic (within-partition) agreement signal.

Figure 2: Synthetic graphs with binary labels $y \in \{0, 1\}$ determined by agreement patterns: (a) across partitions and (b) within partitions. Red edges are informative; others are distractors.

**Synthetic data generation.** We construct a synthetic graph classification dataset where each example consists of a node feature matrix $X \in \mathbb{R}^{N \times d}$ and a binary label $y \in \{0, 1\}$. Each instance contains *two graphs* defined over the same nodes and features: one in which *heterophilic* edges carry the predictive signal and one in which *homophilic* edges carry the signal. Nodes are split into two latent partitions $A$ and $B$ of equal size, and we sample latent signs $z \in \{-1, +1\}^N$ so that $y$ is encoded through pairwise *agreement structure*: (i) cross-partition agreement (heterophily signal; Fig. 2a), and (ii) within-partition agreement (homophily signal; Fig. 2b). For homophilic signal, within each partition we form fixed node pairs and set their signs to match if $y = 1$ and to oppose if $y = 0$. For heterophilic signal, we set $z_B = z_A$ when $y = 1$ and $z_B = -z_A$ when $y = 0$. Node features are generated by placing this sign in a single signal dimension with Gaussian noise and filling the remaining dimensions with independent noise, so $y$ is not trivially recoverable from pooled marginals. The two views differ *only* in which edges expose the relevant relation; the underlying $(X, y)$ is shared. This construction yields two graph structures with different edge semantics over the same instance, making it possible to test whether a view-specific mask can retain semantic edges while suppressing distractors in each view. Finally, we randomly permute node indices independently per graph to prevent shortcut solutions that exploit consistent node ordering.

## 5.1 Learning task-relevant connectivity

We test whether MASKGT can learn a task-appropriate relational inductive bias by selecting which edges should mediate message passing (attention). Concretely, we ask whether the learned mask (i) *improves prediction* on real-world benchmarks where edge semantics are unknown and may vary in usefulness across datasets, and (ii) *denoises* a noisy observed graph by suppressing distractor edges when the ground-truth semantic relations are known.

### 5.1.1 Real-world benchmarks

We evaluate whether MASKGT can learn an effective relational inductive bias on real-world datasets, where edge semantics are not controlled and the usefulness of the given graph structure varies across tasks. We consider both node- and graph-level prediction problems and compare against the unmasked GT as well as standard masking/pruning baselines.

**Data.** We evaluate on both node- and graph-level tasks. For node-classification experiments, we use the WebKB datasets CORNELL, TEXAS, and WISCONSIN, each consisting of a single citation-style graph (Pei et al., 2020). For graph-classification experiments, we use PROTEINS, PTC_MR, MUTAG (Morris et al., 2020), NCI1 (Wale & Karypis, 2006) and OGBG-MOLHIV (Hu et al., 2020). We also include a graph-regression experiment using ZINC (Gómez-Bombarelli et al., 2018). The metric for all datasets except OGBG-MOLHIV and ZINC is accuracy; for OGBG-MOLHIV it is ROC-AUC (higher better), and ZINC is MAE (lower better).

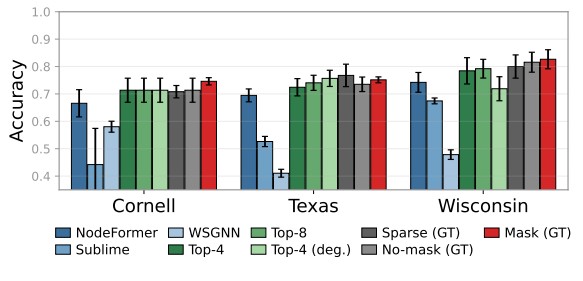

(a) Node classification.

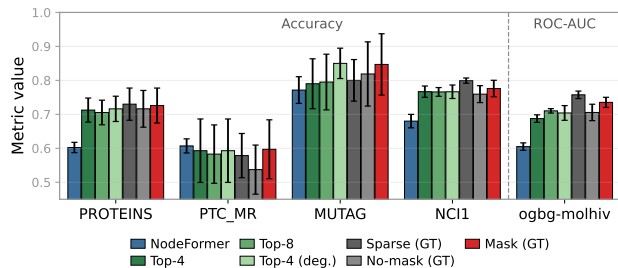

(b) Graph classification

Figure 3: MaskGT improves performance over non-masked GT for both node and graph classification, and outperforms other heursitic-based (green) and GSL (blue) methods.

We use public data splits for all datasets except PROTEINS, PTC_MR, and MUTAG, for which we use random splits of 0.6/0.2/0.2 for training/validation/testing. Further details are provided in Appendix C.

**Baselines.** We compare MASKGT against three types of baselines. (i) Complete/Sparse attention: We include an unmasked GT (NO MASK) with complete attention, and a sparse-attention GT (SPARSE) that restricts attention to edges in the input graph. (ii) Heuristic pruning: We consider top-$k$ pruning by attention weights and a degree-based rule that, for each node, retains the $k$ neighbors with the largest degrees. The degree heuristic preferentially connects nodes to hubs, which can improve global connectivity but may also preserve non-semantic edges. (iii) Graph structure learning (GSL): We include NODEFORMER (Wu et al., 2022), which learns linear-time sparse attention over all node pairs, as well as WSGNN (Lao et al., 2022) and SUBLIME (Liu et al., 2022), which explicitly infer task-relevant structure. WSGNN trains coupled GNNs for structure inference and label prediction, while SUBLIME uses contrastive objectives to align representations from the original and learned graphs. Since WSGNN and SUBLIME are only available for node-level tasks, we omit them for graph classification.

**Set-up.** For GSL methods, we use the implementation provided in OpenGSL (Zhou et al., 2023), as well as code provided from their respective sources. Hyperparameters are determined by following the steps in Appendix D, and we include in Appendix D.1.1. The mask generator module of MaskGT is learned jointly with the GT.

**Results.** Across both node- and graph-level benchmarks, MASKGT (red) generally improves over the dense NO MASK GT baseline and outperforms the heuristic pruning rules and considered GSL baselines (Fig. 3 and Fig. 4a). Wilcoxon p-values are included in Appendix D.2, indicating that the improvements over the no-mask baseline are statistically significant. At the same time, neither sparse attention (SPARSE) nor complete attention (NO MASK) is reliably best across datasets, suggesting that the optimal relational inductive bias is task-dependent. MASKGT provides a learned alternative that adapts the attention support to each task rather than committing to either fixed sparse or fully dense connectivity. We further report the sparsity of

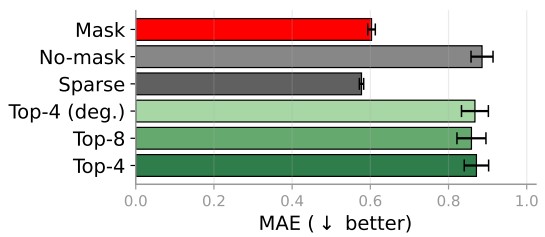

(a) Graph regression (MAE; lower is better)

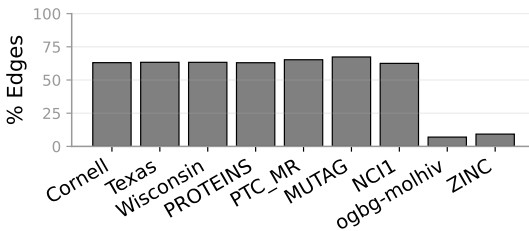

(b) Learned mask sparsity (% of edges retained).

Figure 4: MaskGT improves graph regression and learns sparse, dataset-adaptive connectivity

the learned connectivity in Fig. 4b: MASKGT typically retains about 60% of candidate edges and learns even sparser masks on OGBG-MOLHIV and ZINC.

## 5.2 Generalization across GT architectures

To examine whether MASKGT generalizes beyond a single Graph Transformer architecture, we apply it to two additional GT variants: GPS (Rampášek et al., 2022) and Specformer (SF) (Bo et al., 2023).

**Data.** We use the same experimental set-up as in Sec. 5.1.1. For node classification, we evaluate on the WebKB datasets CORNELL, TEXAS, and WISCONSIN (Pei et al., 2020). For graph classification, we evaluate on PROTEINS, PTC_MR, MUTAG (Morris et al., 2020), NCI1 (Wale & Karypis, 2006), and OGBG-MOLHIV (Hu et al., 2020).

**Results.** Figure 5 compares the no-mask baseline (gray) with MASKGT (red) using GPS and Specformer (SF) as base Graph Transformer architectures. MASKGT improves performance on most evaluated datasets, supporting its use as a modular task-adaptive masking component across different GT backbones.

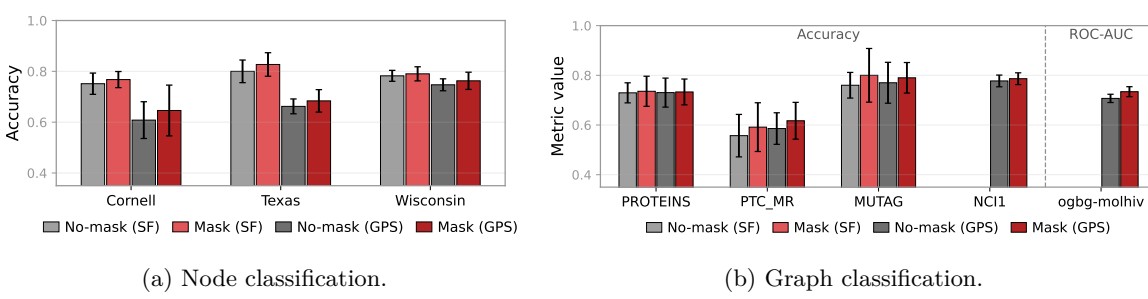

(a) Node classification.  (b) Graph classification.

Figure 5: MASKGT improves performance across different GT architectures: GPS and Specformer. Specformer (SF) did not finish within 6 hours on NCI1 and OGBG-MOLHIV.

### 5.2.1 Controlled denoising

We evaluate whether MASKGT can recover task-relevant connectivity by *filtering* spurious relations from a noisy / corrupted input graph. We consider two settings on the synthetic data: (i) complete (all-to-all) attention: the mask is applied over the fully connected attention graph (as in Fig. 1). (ii) sparse attention: candidate attention edges are restricted to those present in the observed adjacency (plus injected distractors); this forces the learned mask to denoise the edges as it cannot bypass the provided (possibly noisy) adjacency through other edges.

**Data.** We use the synthetic graph-level binary classification dataset and focus on the heterophily-signal regime (Fig. 2a). Labels $y \in \{0, 1\}$ are determined by cross-partition relations between $A$ and $B$, while within-partition edges are non-informative by construction and act as distractors. We control the noise level by increasing the number of injected distractor edges. Each split contains 1000 training graphs, 200 validation graphs, and 200 test graphs.

**Set-up.** We train for 100 epochs with batch size 64. The GT backbone has 2 layers and 2 attention heads, using learning rate $10^{-3}$ and weight decay $5 \times 10^{-4}$. The mask generator module is learned jointly with the GT. We select the best checkpoint by validation performance and report results on the test set.

**Results.** MASKGT consistently improves robustness to noisy edges in both settings. With complete attention, masking compensates for the degradation (Fig. 6a) caused by increasing number of distractor edges, which corrupts the node features via the positional encodings derived from the input graph. With sparse attention, masking explicitly denoises the observed graph: as noise increases, the learned masks retain a higher proportion of signal edges relative to distractors, which is reflected in improved predictive performance (Fig. 6b).

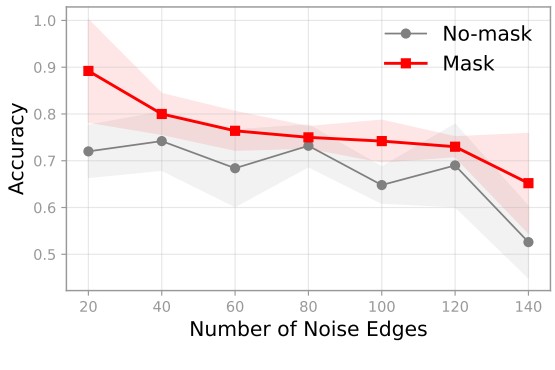
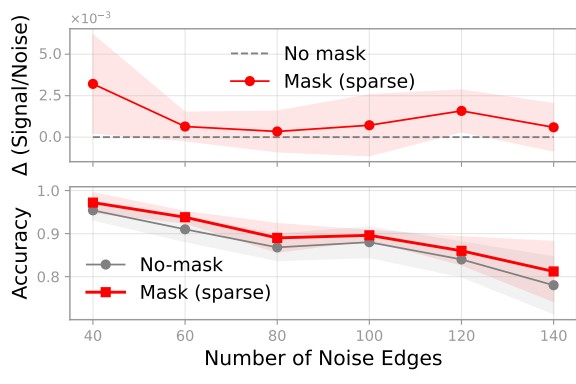

(a) Masking on *complete* attention

(b) Masking on *sparse* attention

Figure 6: MaskGT improves robustness against distractor edges. Denoising on sparse attention highlights that signal-to-noise ratio is improved with masking.

To complement the controlled synthetic study, we also evaluate robustness on real-world graph classification datasets under random structural corruption, where 30% of the original edges are randomly added or deleted. These results are reported in Appendix D.3.

### 5.3 Multi-task learning

We evaluate whether MASKGT improves the adaptivity of a shared GT backbone in a multi-task setting. We train a single base GT across multiple tasks while learning a *task-specific* mask for each task, so that the shared backbone can reuse general representations while the masks bridges between the task and model.

#### 5.3.1 Opposing graph semantics (synthetic)

Graph edges may encode qualitatively different relations across datasets (e.g., homophily vs. heterophily), so a single fixed connectivity prior can be suboptimal. We study an extreme but controlled case with two tasks: one where the predictive signal is carried by *heterophilic* connections and one where it is carried by *homophilic* connections. We train one mask per task while sharing the GT parameters across both tasks. As a baseline, we train the same shared backbone under the same multi-task protocol but *without* any masking.

**Data.** We use the synthetic graph dataset (Fig. 2) and define two tasks: (i) graphs where labels are determined by heterophilic signal edges, and (ii) graphs where labels are determined by homophilic signal edges. Each task contains 1000 training graphs, with 200 graphs for validation and 200 for testing. Each graph has $N = 40$ nodes. To increase the difficulty of each task, we additionally inject distractor edges: for heterophilic-signal graphs we add within-partition Erdős–Rényi distractors with probability $p = 0.3$, while for homophilic-signal graphs we add cross-partition distractors with probability $p = 0.3$.

**Set-up.** We train both tasks jointly by alternating mini-batches between the two datasets while sharing the GT backbone and updating the corresponding task mask for each batch. Further details in Appendix D.1.2.

**Results.** Task-specific masking enables the shared GT backbone to adapt to opposing edge semantics, improving performance on both the heterophilic- and homophilic-signal tasks (Fig. 7a). This suggests that MASKGT can specialize the relational inductive bias per task while reusing a common backbone, whereas a single unmasked model is forced to commit to a single connectivity prior and consequently underperforms when the semantics conflict. This highlight the flexibility of masking for adapting to heterogeneous relational structure across tasks.

#### 5.3.2 Real-world multi-task learning

We extend the multi-task evaluation to real-world graph classification benchmarks to test whether task-specific masking improves adaptivity when datasets differ in domain and edge semantics.

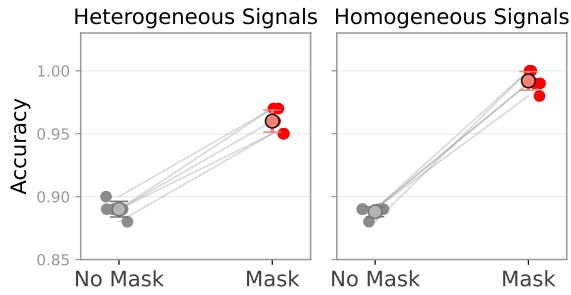
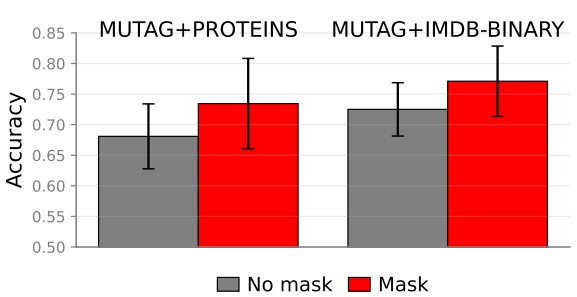

(a) Synthetic graph classification; evaluated separately.

(b) Real-world datasets; evaluated jointly.

Figure 7: Masking improves performance on semantically different tasks.

**Data.** We evaluate two dataset pairs: (i) MUTAG + PROTEINS and (ii) MUTAG + IMDB-BINARY. MUTAG is a molecular graph classification dataset, PROTEINS contains protein graphs for bioinformatics classification, and IMDB-BINARY is a graph classification benchmark derived from movie co-appearance/collaboration networks. These combinations span distinct domains and relational meanings, making them a natural testbed for task-adaptive inductive bias.

**Set-up.** We train for 20 epochs with 50 optimization steps per epoch. All shared-backbone hyperparameters (e.g., learning rate, weight decay) are kept identical across the two tasks in each pair (Appendix D.1.2.). At each step, we sample which dataset provides the next mini-batch using size-power sampling with $\alpha = 0.5$, i.e., $p(d) \propto |D_d|^{0.5}$ with a minimum probability of 0.05. We share the GT backbone across tasks, while using a dataset-specific mask module and prediction head for each dataset. We report test performance on both datasets.

**Results.** Task-specific masking improves the ability of the shared GT to accommodate heterogeneous datasets. Compared to a strong baseline that shares the backbone but uses task-specific heads without masking, MASKGT results in larger gains across both dataset pairs (Fig. 7b), indicating that adapting connectivity—not only the readout—is important for effective multi-task transfer on graphs.

### 5.4 Transfer learning

We evaluate MASKGT in a transfer setting where adaptivity is constrained: the pretrained backbone is frozen, and only lightweight task-specific components are trained for the downstream task. This isolates whether learning a mask can effectively adapt the relational inductive bias *without* updating the base GT parameters.

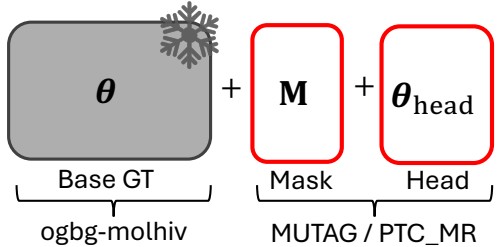
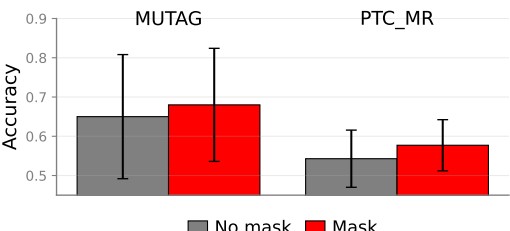

(a) Transfer setup: freeze pretrained GT backbone and fine-une only mask and task head.

(b) Transfer performance on MUTAG and PTC_MR is improved by masking over adapting the head alone (No mask).

Figure 8: MaskGT enables parameter-efficient transfer. With the pretrained GT backbone frozen and fine-tune only the mask and the head, leads to improved accuracy on MUTAG and PTC_MR.

**Data.** We pretrain the GT backbone on OGBG-MOLHIV and transfer to MUTAG and PTC_MR. OGBG-MOLHIV contains molecular graphs labeled by HIV activity. MUTAG and PTC_MR are molecule classification datasets where the labels indicate whether a compound is mutagenic and carcinogenic, respectively.

**Set-up.** We first train the base GT on OGBG-MOLHIV *without* masking. During transfer, we freeze all backbone parameters and fine-tune only (i) a task-specific prediction head and (ii) the mask module (or no mask for the baseline), as illustrated in Fig. 8a. Thus, any downstream improvements reflect the ability of masking to reweight and prune interactions in the frozen attention connectivity rather than additional backbone capacity. Hyperparameters are included in Appendix D.1.3.

**Results.** Learning a task-specific mask consistently improves fine-tuned performance over the head-only baseline (Fig. 8b), indicating that adapting connectivity is an effective and parameter-efficient mechanism to transfer models for graphs.

### 5.5 Mask hyperparameter ablations

We examine two MASKGT hyperparameters: the Gumbel-Softmax temperature $\tau$ and whether the mask is shared across attention heads or learned separately for each head. We evaluate both ablations on the synthetic dataset, CORNELL for node classification, and OGBG-MOLHIV for graph classification, reporting results over 5 runs.

**Gumbel-Softmax temperature.** The temperature $\tau$ controls the sharpness of the mask relaxation. Larger values yield smoother mask probabilities, while smaller values make the relaxation closer to a hard binary gate. As shown in Fig. 9a, the optimal temperature varies across datasets, but performance remains stable over a moderate range. Although very small values of $\tau$ degrade performance, suggesting that overly sharp relaxations can introduce training instability.

**Per-head vs. shared mask.** We compare learning a separate mask for each attention head against sharing a single mask across heads. Per-head masks allow different heads to capture distinct interaction patterns, while a shared mask imposes a stronger common constraint. As shown in Fig. 9b, the better choice is dataset-dependent: the synthetic dataset and CORNELL benefit from the additional flexibility of per-head masking, whereas OGBG-MOLHIV benefits from the regularization of a shared mask.

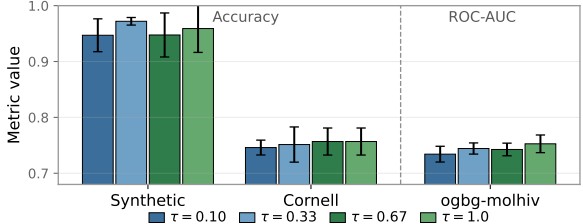 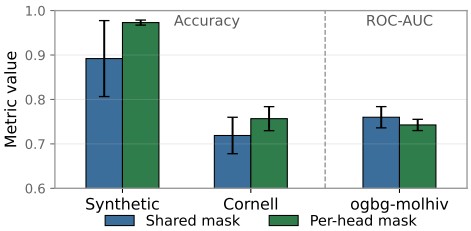

(a) Ablation on the Gumbel-Softmax temperature $\tau$.     (b) Ablation on shared vs. per-head masking.

Figure 9: Ablation of key masking hyperparameters on synthetic, CORNELL, and OGBG-MOLHIV data over 5 runs with mean and standard deviation.

### 5.6 Discussion

Across experiments, MASKGT learns task-relevant connectivity: in controlled denoising, it suppresses distractor edges and retains informative relations, improving the effective signal-to-noise ratio; on real node- and graph-level benchmarks, it consistently improves the same GT backbone, outperforming heuristic pruning, representative GSL baselines, and fixed complete/sparse attention variants (Fig. 3). Beyond single-task gains, MASKGT improves adaptivity under changing edge semantics by learning distinct masks per task in multi-task learning (Fig. 7) and by enabling better transfer adaptation of a pre-trained GT model (Fig. 8b). We report compute and memory in Appendix D.4; a key limitation shared with GTs is scalability, since

learning dense pairwise gates adds $O(n^2)$ cost in the number of nodes, motivating future work on restricting candidate pairs or using structured mask parameterizations for large graphs.

## 6 Conclusion

We introduced MASKGT, a simple and GT-agnostic module that learns *task-adaptive connectivity* by gating attention edges with a discrete, sparse mask. Rather than fixing attention to the input topology or defaulting to dense all-to-all interactions, MASKGT learns which node pairs should communicate *within* self-attention, resulting in a a task-adaptive relational inductive bias. Across controlled synthetic tests and real-world benchmarks, MASKGT suppresses spurious interactions to denoise corrupted graphs, improves predictive performance and robustness over strong GT baselines, and supports parameter-efficient multi-task and transfer by localizing task-specific structure in the mask while reusing a shared backbone. These results suggest that learning connectivity as a lightweight, modular component is a practical path toward general-purpose graph models that can specialize across diverse domains and edge semantics.

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

## A  Motivation

In this work, we argue that the *interaction structure* in a graph model should be treated as a learnable, task-dependent component. In this appendix we provide two additional motivating observations: (i) even a simple scalar control of edge bias can yield markedly different optima across datasets, suggesting that the "right" relational bias is not universal and is task-dependent; and (ii) GNNs and GTs differ primarily in the *support* of interactions, so learning attention support provides a principled way to interpolate between message passing and global attention. Together, these motivate MASKGT, which learns a discrete attention support inside the transformer rather than hand-designing sparsity or selecting it by hyperparameter search.

### A.1  Optimal relational bias is task dependent

In Kreuzer et al. (2021), they introduced a hyperparameter $\gamma$ to up/down weight the edges in the graph in the attention as follows:

$$
w_{ij}^{k,l} = \begin{cases} \frac{1}{1+\gamma} \, \mathrm{softmax}_j\Big(\frac{(\mathbf{W}_Q^{k,l}\mathbf{h}_i^{(l-1)})^\top(\mathbf{W}_K^{k,l}\mathbf{h}_j^{(l-1)})}{\sqrt{d_k}}\Big) & \text{if } (i,j) \in \mathcal{E}, \\ \frac{\gamma}{1+\gamma} \, \mathrm{softmax}_j\Big(\frac{(\mathbf{W}_Q^{k,l}\mathbf{h}_i^{(l-1)})^\top(\mathbf{W}_K^{k,l}\mathbf{h}_j^{(l-1)})}{\sqrt{d_k}}\Big) & \text{if } (i,j) \notin \mathcal{E}. \end{cases}
$$

When $\gamma$ is large, the model more closely resembles a GT; when it is small, it more closely resembles a GNN. As seen in Fig. 10, the optimal $\gamma$ varies across dataset: ZINC prefers stronger reliance on input edges, whereas CLUSTER improves with weaker edge bias (Kreuzer et al., 2021) . Intuitively, ZINC is a molecular dataset where local chemistry (e.g., bonds and cycles) is predictive, while CLUSTER is generated from a stochastic block model (SBM) (Abbe, 2018), where the observed edges may be less directly aligned with the target labels. This shows that different datasets/tasks benefit from different level of bias towards the sparse topology vs. the all-to-all interaction. While $\gamma$ interpolates between edge-biased and global attention, it is a *global* knob: it cannot remove specific spurious edges or add missing task-relevant interactions, which motivates learning the pairwise interaction mask.

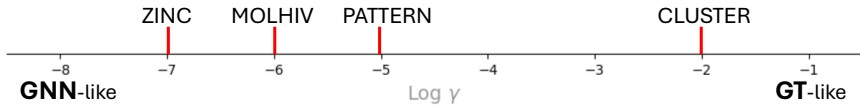

Figure 10: Optimal performance varies across datasets with the strength of relational inductive bias.

### A.2  GNN as a special case of GT

Graph Neural Networks (GNNs) can be viewed as special cases of GT under appropriate choices of connectivity and attention weights. In particular, Joshi (2025) formalize that a standard Transformer (Vaswani et al., 2017) corresponds to a message-passing GNN operating on a fully connected graph, i.e., $\mathcal{N}(i) = \mathcal{V}$ for all $i \in \mathcal{V}$.

Starting from the GNN operation in Sec. 2.1:

$$\mathbf{m}_{ij}^{(l-1)} = \mathrm{MESSAGE}^{(l)}(\mathbf{h}_i^{l-1}, \mathbf{h}_j^{l-1}), \forall j \in \mathcal{N}(i) \tag{5}$$

$$\mathbf{m}_{\mathcal{N}(i)}^{(l-1)} = \mathrm{AGGREGATE}^{(l)}(\{\mathbf{m}_{ij}^{(l-1)}, \forall j \in \mathcal{N}(i)\}) \tag{6}$$

$$\mathbf{h}_i^{(l)} = \mathrm{UPDATE}^{(l)}(\mathbf{h}_i^{(l-1)}, \mathbf{m}_{\mathcal{N}(i)}^{(l-1)}). \tag{7}$$

We can rewrite these operations in terms of single-head self-attention in GT, replacing the sparse topology-based support $\mathcal{C}(i) = \mathcal{N}(i)$, with dense self-attention $\mathcal{C}(i) = \mathcal{V}$:

$$
\begin{aligned}
\mathbf{m}_{ij}^{(l-1)} &= \text{MESSAGE}^{(l)}(\mathbf{h}_i^{l-1}, \mathbf{h}_j^{l-1}), \forall j \in \mathcal{V} \\
&= a_{ij}^l \mathbf{W}_V^l \mathbf{h}_j^{(l-1)}, \forall j \in \mathcal{V} \text{ and } a_{ij}^l \text{ as in Eq. 2} \\
\mathbf{m}_{\mathcal{V}}^{(l-1)} &= \text{AGGREGATE}^{(l)}(\{\mathbf{m}_{ij}^{(l-1)}, \forall j \in \mathcal{V}\}) \\
&= \sum_{j \in \mathcal{V}} \mathbf{m}_{ij}^{(l-1)} \\
\mathbf{h}_i^{(l)} &= \text{UPDATE}^{(l)}(\mathbf{h}_i^{(l-1)}, \mathbf{m}_{\mathcal{V}}^{(l-1)}) \\
&= \mathbf{W}_O^l \mathbf{m}_{\mathcal{V}}^{(l-1)}.
\end{aligned}
$$

Here, $a_{ij}^l$ denotes the attention weight defined in Eq. 2. Substituting these definitions recovers the same update as Eq. 3. This equivalence highlights that GNNs and GTs differ mainly in their interaction *support*; MASKGT leverages this view by learning the support (via $\mathbf{M}$) inside attention.

## B    MaskGT

### B.1    Gumbel-Softmax (Concrete) relaxation and Straight-Through Estimator (STE)

We use the Gumbel-Softmax trick to sample the discrete attention mask $\mathbf{M}$ while allowing gradient to flow. The trick uses the concrete distribution to relax discrete random variables. The relaxed variable of $\mathbf{M}_{ij}$ (corresponding to the edge between node $i$ and $j$) is denoted as $\mathbf{M}_{ij}^r \in (0, 1)$, with probability distribution given by the binary Concrete distribution (from Maddison et al. (2017, App. B)):

$$
\mathbf{M}_{ij}^r = \frac{1}{1 + \exp(-\frac{\log \alpha_{ij} + L}{\tau})}, \qquad p(\mathbf{M}_{ij}^r(\pi_{ij})) = \frac{\tau \alpha (\mathbf{M}_{ij}^r(\pi_{ij}))^{-\tau-1}(1 - (\mathbf{M}_{ij}^r(\pi_{ij}))^{-\tau-1}}{(\alpha(\mathbf{M}_{ij}^r(\pi_{ij}))^{-\tau}(1 - (\mathbf{M}_{ij}^r(\pi_{ij}))^{-\tau})^2}. \tag{8}
$$

where $\alpha_{ij} = \frac{\pi_{ij}}{1 - \pi_{ij}}$ and $L = \log \frac{\epsilon_{ij}}{1 - \epsilon_{ij}}$ is a sample from the Logistic distribution, with $\epsilon_{ij} \sim \mathcal{U}(0, 1)$. The temperature $\tau$ controls the degree of relaxation (we fixed $\tau = 0.67$). To make the relaxed $\mathbf{M}_{ij}^r$ fully discrete, we use straight-through estimator (STE) (Bengio et al., 2013) which thresholds $\mathbf{M}_{ij}^r$ to make it binary in the forward pass, while the gradient is calculated using the continuous $\mathbf{M}_{ij}^r$. In the forward pass, the mask $\mathbf{M}$ is sampled from the scores outputted by the mask generator: $\mathbf{S}$.

## C    Datasets

We provide a summary of the datasets used in the experiments in Table 1. Cornell, Texas, and Wisconsin are WebKB node-classification graphs (Pei et al., 2020): nodes represent university web pages, edges represent hyperlinks, and node features are typically bag-of-words page representations. In contrast, PROTEINS, PTC_MR, MUTAG, and IMDB-BINARY are graph-classification datasets from the TU Dortmund collection (Morris et al., 2020). PROTEINS contains graphs derived from protein structures, PTC_MR and MUTAG are small-molecule datasets with binary labels (PTC_MR corresponds to carcinogenicity for male rats, and MUTAG is a mutagenicity benchmark), NCI1 is a molecular graph dataset derived from National Cancer Institute anti-cancer screening data, and IMDB-BINARY is a social-network dataset in which each graph represents an ego-network from IMDb with a binary label. Similarly, ogbg-molhiv is an OGB molecular property prediction benchmark (Hu et al., 2020), where graphs represent molecules (atoms as nodes and bonds as edges) and labels indicate HIV activity. For graph regression, we include ZINC which consists of molecular graph and the task is to predict solubility. We use public data splits for all the datasets except PROTEINS, PTC_MR, and MUTAG, for which we use random splits of 0.6/0.2/0.2 for training/validation/testing.

Table 1: Dataset statistics. Average numbers of nodes and edges are reported for graph-level tasks.

| Dataset | # Graphs | # Nodes | # Edges | # Classes |
|---------|----------|---------|---------|-----------|
| Cornell | 1 | 183 | 298 | 5 |
| Texas | 1 | 183 | 325 | 5 |
| Wisconsin | 1 | 251 | 515 | 5 |
| PROTEINS | 1,113 | 39.06 | 72.82 | 2 |
| PTC_MR | 344 | 14.29 | 14.69 | 2 |
| MUTAG | 188 | 17.93 | 19.79 | 2 |
| IMDB-BINARY | 1,000 | 19.77 | 96.53 | 2 |
| NCI1 | 4,110 | 29.8 | 32.3 | 2 |
| ogbg-molhiv | 41,127 | 25.5 | 27.5 | 2 |
| ZINC | 249,456 | 23.2 | 49.8 | N/A |

# D    Experiment

We provide our code as supplementary materials, and will be made publicly available upon acceptance.

## D.1    Set-up and Hyperparameters

We conducted a random search over learning rate (log-uniform in $[10^{-4}, 3 \times 10^{-2}]$), weight decay (log-uniform in $[10^{-6}, 10^{-2}]$), dropout (uniform in $[0.0, 0.6]$), attention dropout (uniform in $[0.0, 0.3]$), hidden dimension ($\{64, 128, 256\}$), number of layers ($\{2, 3, 4\}$), and number of attention heads ($\{2, 4, 8\}$), resampling heads until the hidden dimension is divisible by the number of heads. We provide a budget of 50 trials with 100 epochs each, and model is selected based on performance on validation set.

**Mask generator** The mask generator module $g_{\boldsymbol{\Phi}}$ of MASKGT can be defined as any model that outputs the scores for each pair of edges $\mathbf{S} \in \mathbb{R}^{n \times n}$. In our experiment, we use 2-layer MLP with ReLU activation. We only allow mask to be varied across attention heads and not across layers. The generator predicts a scalar score $\mathbf{S}_{ij}$ for nodes $i$ and $j$ by taking as input concatenation of: $[\tilde{h}_i^0, \tilde{h}_j^0, \tilde{h}_i^0 - \tilde{h}_j^0, \tilde{h}_i^0 * \tilde{h}_j^0]$, where $\tilde{h}_i^0$ is the input features for node $i$ with positional encodings. We learn the mask generator using the Adam optimizer, using the same larning rate and weight decay as the GT.

**MaskGT-specific hyperparameters.** The only hyperparameter we vary in MASKGT is the size of the hidden layer, and whether to mask per-head or use the same mask over all attention heads.

### D.1.1    Real-world benchmark experiments

Here, we provide the hyperparameters used in Sec. 5.1.1

Table 2: Hyperparameters used for MaskGT and No-mask GT.

| Dataset | L | H | Dim | LR | WD | MaskGT Dim | MaskGT per-head |
|---------|---|---|-----|-----|-----|------------|-----------------|
| Cornell | 4 | 4 | 128 | 0.0007 | 0.00001 | 64 | True |
| Texas | 4 | 2 | 256 | 0.002 | 0.008 | 32 | True |
| Wisconsin | 4 | 2 | 256 | 0.002 | 0.008 | 64 | False |
| PROTEINS | 4 | 4 | 128 | 0.001 | 0.0005 | 128 | True |
| PTC_MR | 8 | 4 | 128 | 0.0001 | 0.0005 | 32 | False |
| MUTAG | 3 | 4 | 128 | 0.001 | 0.0005 | 64 | True |
| ogbg-molhiv | 2 | 8 | 256 | 0.0001 | 0.00001 | 64 | False |

### D.1.2 Multi-task learning.

**Synthethic dataset.** On the synthetic dataset, we use a learning rate of 0.001, weight decay of 0.0005, hidden dimension of 128, 4 layers, 4 attention heads, and dropout of 0.2. For MASKGT, the mask generator module has hidden dimension 128 and is not learned per-head (one mask for the model). We trained for 200 epochs for each task.

**Real-world datasets.** For MUTAG and PROTEINS, we use a learning rate of 0.001, weight decay of 0.0005, hidden dimension of 128, 4 layers, 4 attention heads, and dropout of 0.2; for MASKGT, the mask generator has hidden dimension 64 and is not learned per-head (i.e., one mask is shared across layers and heads).

For MUTAG and IMDB-BINARY, we use a learning rate of 0.0001, weight decay of 0.00001, hidden dimension of 128, 3 layers, 4 attention heads, and dropout of 0.2; for the mask generator, we use hidden dimension 64 and learn the mask per-head.

### D.1.3 Transfer learning.

The model is pre-trained on OGBG-MOLHIV with hidden dimension 128, 4 layers, 4 attention heads, dropout of 0.1, learning rate of 0.0001, weight decay of 0.00001, and 200 training epochs.

Then, we fine-tune on MUTAG for 100 epochs with a learning rate of 0.0001 and head learning rate of 0.001; the mask module uses hidden dimension 32 and learns a single shared mask. We also fine-tune on PTC_MR for 100 epochs with a learning rate of 0.0001 and head learning rate of 0.001; the mask module uses hidden dimension 16 and is learned per-head.

### D.2 Wilcoxon p-value

We report Wilcoxon signed-rank test p-values for the results in Fig. 3 from Sec. 5.1.1. The test compares MASKGT with the corresponding complete-attention no-mask baseline across random seeds, using the GT backbone (Dwivedi & Bresson, 2021). All datasets except MUTAG reach the conventional 0.05 significance threshold, supporting that the observed gains are generally consistent across random seeds.

Table 3: Wilcoxon p-values comparing MASKGT against the no-mask baseline in Fig. 3.

| Dataset | Cornell | Texas | Wisconsin | PROTEINS | PTC_MR | MUTAG | NCI1 | ogbg-molhiv |
|---|---|---|---|---|---|---|---|---|
| GT | 0.04688 | 0.03931 | 0.03564 | 0.04925 | 0.01562 | 0.05532 | 0.02734 | 0.01562 |

### D.3 Adversarial random corruption

We evaluate the robustness of MaskGT under structural corruption on real-world graph classification datasets: PROTEINS, PTC_MR, MUTAG, NCI1, and OGBG-MOLHIV. We consider two Graph Transformer backbones: the vanilla Graph Transformer (GT) (Dwivedi & Bresson, 2021) and GPS (Rampášek et al., 2022).

**Set-up.** We consider two types of random structural corruption: edge addition and edge deletion. For edge addition, we randomly add edges equal to 30% of the original edge count. For edge deletion, we randomly remove 30% of the original edges. For each backbone, we compare complete attention without masking against MaskGT, and repeat each experiment over 10 runs.

**Results.** Figure 11 shows the performance of Graph Transformer and GPS under random edge addition and deletion. MASKGT generally improves robustness over the corresponding no-mask baseline across both backbones and corruption types. These results indicate that learned masking can mitigate the effect of structural perturbations by suppressing less useful communication paths and retaining task-relevant interactions.

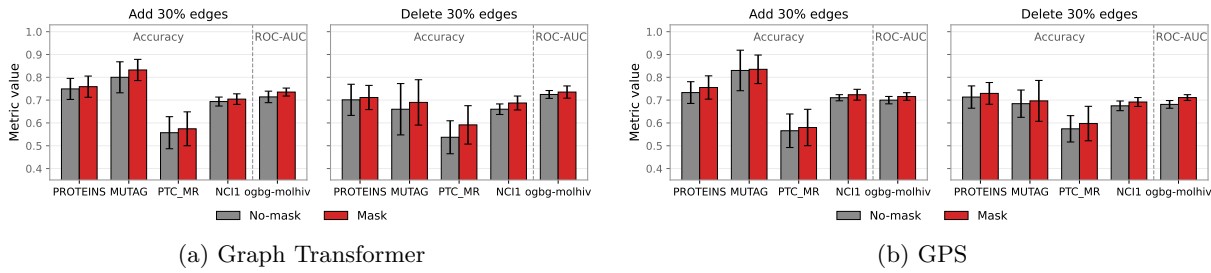

(a) Graph Transformer                              (b) GPS

Figure 11: Performance under 30% random edge addition and deletion, averaged over 10 runs. MaskGT generally improves over the no-mask baseline for both Graph Transformer and GPS.

## D.4    Computation cost

We report the computational overhead of MASKGT in terms of training time and GPU memory. For graph classification on PTC_MR, training without masking takes 59.13s and uses 782MB of GPU memory, while MASKGT takes 61.17s and uses 851MB. For node classification on TEXAS, training without masking takes 6.58s and uses 563MB, while MASKGT takes 13.91s and uses 763MB. The larger increase in computational cost on node-level tasks is expected because there is a single large graph, and the mask generator predicts an $n \times n$ score matrix (where $n$ is the number of nodes). For graph-level tasks, where graphs are relatively small, the computational overhead is lower.

