# OpenReview forum: "MaskGT: Learning Task-Adaptive Connectivity in Graph Transformers"
_TMLR — Decision pending for TMLR_

### Review · Reviewer_unRn · 2026-03-25

**Summary Of Contributions:**

This paper proposes MaskGT, which aims to learn a task-adaptive mask for graph transformers (GT). MaskGT is agnostic to different GT architectures and learns a discrete, sparse gate through Gumbel-softmax. Experiments on both real and synthetic datasets validate the effectiveness of MaskGT.

Pros:

1. This paper is well-motivated. GTs are good at learning global information but ignore the local connectivity. Learning a sparse gate for self-attention can address this issue.

2. This paper makes diverse experiments to validate the effectiveness of MaskGT, including classification, regression, denoising, multi-task learning, and transfer learning.

**Audience:**

Yes

**Audience Explanation:**

The sparsity of self-attention is important. The design of MaskGT may improve the interpretability and robustness of GTs.

**Claims And Evidence:**

No

**Claims Explanation:**

1. This paper claims that MaskGT is agnostic to GT architectures. However, the experiments only adopt NodeFormer as the backbone. It would be better if the authors could evaluate more GT architectures, such as Graph-VIT [1], GPS [2], and Specformer [3].

2. This paper claims that MaskGT can improve the accuracy and robustness of GTs. However, this approach is not convincing due to the lack of experiments on adversarial attacks.

3. The performance improvement of MaskGT is marginal in Figure 3. For example, it is surpassed by the sparse architecture in the Texas and ogbg-molhiv datasets, indicating that MaskGT cannot learn ground-truth structures.

[1] A Generalization of ViT/MLP-Mixer to Graphs.

[2] Recipe for a General, Powerful, Scalable Graph Transformer.

[3] Specformer: Spectral Graph Neural Networks Meet Transformers.

**Requested Changes:**

The authors should add additional experiments on various GT architectures and provide more results to backup their claims.

---

> ### Author Response · Authors · 2026-06-01
>
> We thank the reviewer for their assessment and constructive feedback. We have revised the paper (in green) to strengthen the evidence for GT-agnostic applicability, evaluate robustness under structural perturbations, improve statistical reliability, and add further ablations.
>
> 1. **Experiments with additional GT backbones.**
>    To better support the GT-agnostic claim, we added experiments in Sec. 5.2 using two additional Graph Transformer backbones: Specformer [1] and GPS [2]. The results show that MaskGT improves performance across different GT architectures on both node- and graph-classification tasks, supporting its use as a modular masking component.
>
> 2. **Robustness under structural perturbations.**
>    To further evaluate robustness, we added adversarial random corruption experiments in Appendix D.3, where 30% of the original edges are randomly added or deleted. We evaluate both vanilla GT and GPS backbones. The results show that MaskGT improves robustness under structural perturbations, supporting the claim that learned masking can suppress spurious communication channels.
>
> 3. **Statistical reliability and marginal improvements.**
>    We increased the number of repeated runs from 5 to 10 and expanded the graph-classification evaluation by adding NCI1, a molecular benchmark derived from the National Cancer Institute anti-cancer screening data, as shown in the updated Fig. 3. We also added p-value analyses in Appendix D.2, which indicate that MaskGT’s improvements over the no-mask baseline are statistically significant across random seeds. We agree that MaskGT does not outperform every sparse architecture on every dataset. The goal of MaskGT is not to recover a unique ground-truth structure, but to learn a task-adaptive attention support. A fixed sparse architecture can be competitive when its inductive bias aligns well with the task, as seen in Texas and ogbg-molhiv. In contrast, MaskGT provides a learned alternative that adapts the communication structure to the task and improves over dense attention in the overall results.
>
> 4. **Additional ablations.**
>    We added ablation studies in Sec. 5.5 on the Gumbel-Softmax temperature and per-head masking design to clarify the effect of key hyperparameters.
>
> We hope these revisions address the reviewer’s concerns and would appreciate reconsideration of the assessment.
>
>
> [1] Deyu Bo, Chuan Shi, Lele Wang, and Renjie Liao. Specformer: Spectral Graph Neural Networks Meet Transformers. In *International Conference on Learning Representations*, 2023.
>
> [2] Ladislav Rampášek, Mikhail Galkin, Vijay Prakash Dwivedi, Anh Tuan Luu, Guy Wolf, and Dominique Beaini. Recipe for a General, Powerful, Scalable Graph Transformer. In *Advances in Neural Information Processing Systems*, 2022

---

### Review · Reviewer_3G28 · 2026-05-18

**Summary Of Contributions:**

The paper introduces MaskGT, a module designed to enhance Graph Transformers (GTs) by learning a discrete sparse gate over attention edges. This approach allows the model to adaptively determine which node pairs can communicate within self-attention, injecting a task-specific relational inductive bias without fully relying on the input adjacency. The authors demonstrate that MaskGT improves performance and robustness by suppressing spurious interactions under structural noise, while also enabling parameter-efficient multi-task and transfer learning by localizing task-specific structure in the mask.
While the contributions are technically sound, they appear incremental, as the core idea revolves around adding a masking layer to existing GT architectures. Moreover, the motivation for explicitly modeling the mask (rather than relying on the attention mechanism to implicitly learn it) remains unclear, and the paper does not fully justify why this approach is superior to letting the attention scores naturally suppress irrelevant edges.

**Audience:**

Yes

**Audience Explanation:**

Some individuals in TMLR’s audience would likely find the findings relevant, particularly those working on graph representation learning, self-attention mechanisms, or adaptive model architectures. The idea of introducing a task-adaptive inductive bias to GTs is timely and aligns with ongoing efforts to make graph models more flexible and interpretable. Researchers interested in balancing computational efficiency with expressive power may also appreciate the potential of MaskGT to suppress spurious interactions and improve robustness. However, the incremental nature of the contribution and the limited experimental validation may reduce its broader appeal.

**Broader Impact Concerns:**

The paper does not raise significant ethical concerns that would necessitate a Broader Impact Statement. Nevertheless, I would suggest the authors to briefly discuss whether the learned masks could inadvertently encode or amplify biases present in the training data for example in social network analysis.

**Claims And Evidence:**

No

**Claims Explanation:**

The claims about MaskGT’s performance and robustness improvements are not fully supported by the experimental evidence provided. The high standard deviation in the results, combined with the use of only five random runs, makes it difficult to conclusively determine whether MaskGT outperforms baselines like the No-mask approach. In several cases, the overlap in error bars suggests that MaskGT may perform similarly to its attention-based counterparts, undermining the authors’ assertions of superiority. Additionally, the inconsistency in evaluation metrics (such as using a different performance measure for the ogbg-molhiv dataset compared to other graph classification tasks) raises questions about the fairness and comparability of the results.

**Requested Changes:**

The following issues should be addressed to improve the paper:
- Increase the number of random runs from 5 to at least 20 to reduce the standard deviation and ensure the statistical significance of the results. The current variability makes it impossible to confidently assert MaskGT’s superiority over baselines.
- Expand the experimental evaluation to include a broader range of baselines and more challenging datasets. The current evaluation is too limited to demonstrate the generality and effectiveness of MaskGT compared to state-of-the-art methods.
- Address the inconsistency in evaluation metrics by standardizing the use of either accuracy or AUC-ROC across all graph classification datasets. This is essential for fair comparisons and reproducibility.
- Provide a more thorough ablation study to analyze the impact of hyperparameters, such as the temperature of the Gumbel softmax (e.g., 0.67), on the model’s performance. The arbitrary selection of these values without justification weakens the paper’s credibility.
- Clarify the motivation for explicitly modeling the mask. The authors should address why the attention mechanism alone cannot learn to suppress irrelevant edges and how MaskGT provides a distinct advantage in this regard.

---

> ### Author Response · Authors · 2026-06-01
>
> We thank the reviewer for the constructive feedback. We have revised the paper (in green) to strengthen the empirical evidence, clarify the evaluation protocol, and better motivate explicit mask learning.
>
> 1. **Statistical reliability.**
>    We increased the number of repeated runs from 5 to 10 and added statistical significance tests for the main graph-classification results in Appendix D.2. The p-values show that MaskGT’s improvements over the no-mask baseline are statistically significant across random seeds. We also added NCI1, a molecular benchmark derived from the National Cancer Institute anti-cancer screening data, to the graph-classification evaluation in Fig. 3.
>
> 2. **Broader evaluation.**
>    To support the GT-agnostic claim, we implemented MaskGT on two additional Graph Transformer backbones, Specformer [1] and GPS [2]. The results in Sec. 5.2 show that MaskGT improves performance across different GT architectures on both node- and graph-classification tasks. We also added an adversarial random corruption experiment in Appendix D.3, where 30% of the original edges are randomly added or deleted. The results show that MaskGT improves robustness under structural perturbations.
>
> 3. **Metric clarification.**
>    We follow the official OGB protocol by reporting ROC-AUC for ogbg-molhiv, and report accuracy for the other graph-classification datasets following standard benchmark practice. We revised Fig. 3 and the corresponding text to make this distinction explicit.
>
> 4. **Ablation studies.**
>    We added ablations in Sec. 5.5 on the Gumbel-Softmax temperature and the per-head masking design. These results show that MaskGT is stable over a moderate temperature range and clarify the effect of using more flexible masks.
>
> 5. **Why explicit masking?**
>    We revised Sec. 3 to more clearly frame masking as structural regularization. By learning a binary support, MaskGT restricts information flow to a sparse communication graph, constraining the hypothesis space of the attention to $\mathcal{H}\_\text{mask} \subset \mathcal{H}\_\text{dense}$ . The mask constrains the relational hypothesis space by selecting which node pairs may communicate, while attention learns how strongly to weight those permitted interactions.
>
> We hope these revisions address the reviewer’s concerns. If the reviewer finds the changes satisfactory, we would appreciate reconsideration of the assessment.
>
> [1] Deyu Bo, Chuan Shi, Lele Wang, and Renjie Liao. Specformer: Spectral Graph Neural Networks Meet Transformers. In *International Conference on Learning Representations*, 2023.
>
> [2] Ladislav Rampášek, Mikhail Galkin, Vijay Prakash Dwivedi, Anh Tuan Luu, Guy Wolf, and Dominique Beaini. Recipe for a General, Powerful, Scalable Graph Transformer. In *Advances in Neural Information Processing Systems*, 2022

---

### Review · Reviewer_HwXy · 2026-05-21

**Summary Of Contributions:**

This paper proposes MaskGT, a lightweight GT-agnostic module that achieves task adaptive attention connectivity by learning discrete binary masks on self-attention edges. The core idea of the paper is to use attention support as a learnable component, thereby injecting learnable sparse relation inductive bias into global attention, without relying entirely on input adjacency matrix or forcing full connectivity.

The paper is well organized and clearly written. The proposed method is insightful and adaptive, while the experiments in this paper indicate MaskGT's robustness and adaptivity.
However, the experimental evidences for the generalizability and scalability of the method is insufficient, and the proposed GT-agnostic characteristic also lacks sufficient analysis.

**Audience:**

Yes

**Audience Explanation:**

TMLR readers who are interested in graph machine learning, Transformer variants, and parameter efficient fine-tuning (PEFT) will be interested in this paper.

**Broader Impact Concerns:**

This work focuses on optimizing the basic algorithm of deep graph learning, without involving sensitive privacy data, the risk of bias against specific groups, and potential significant negative social impacts. Therefore, there is no need of additional Broad Impact Statement.

**Claims And Evidence:**

Yes

**Claims Explanation:**

The main contributions of the method proposed in the paper, especially "task-adaptive connectivity in attention" and "modular adaptation", have been reliably validated through experiments on the synthetic benchmarks and multi-task / transfer learning tasks.
However, the core claim of "GT-agnostic" has almost no direct evidence. Meanwhile, the experiment lacks large-graph evaluations with an average node count greater than 50. These make the "more general" claim in the paper need to further verification.

**Requested Changes:**

GT-agnostic evidence: The paper or its appendix should include experiments of at least one different architecture to support this claim.

More ablation studies: Add mask generator ablation, necessity of per-head / per-layer, and temperature sensitivity to enhance readers' understanding of the proposed method.

Expand dataset and experimental scale: It suggests adding experiments on large-graph dataset (e.g., ogbn-arxiv, or ogbn-products) to enhance the argumentation of generalizability.

---

> ### Author Response · Authors · 2026-06-01
>
> We thank the reviewer for the constructive suggestions. We have revised the paper (in green) to strengthen the empirical support for MaskGT’s generality, clarify its architectural scope, and expand the ablation analysis.
>
> 1. **Evidence for GT-agnostic applicability.**
>    We added a new set of experiments in Sec. 5.2 to evaluate MaskGT on two additional Graph Transformer architectures, SpecFormer and GPS, across both node- and graph-classification datasets. These results support the GT-agnostic design of MaskGT, showing that the same masking mechanism can improve different Graph Transformer backbones.
>
> 2. **Additional ablation studies.**
>    We added new ablations in Sec. 5.5 to study the effect of the Gumbel-Softmax temperature and the per-head masking design. These experiments clarify when fine-grained mask parameterization is beneficial and show that MaskGT remains stable across a moderate range of temperature values.
>
> 3. **Expanded experimental scale and robustness analysis.**
>    While very large node-level benchmarks such as ogbn-arxiv and ogbn-products remain challenging for many full-attention Graph Transformers due to scalability constraints, we expanded our evaluation on graph-classification tasks by adding NCI1, a molecular benchmark derived from the National Cancer Institute anti-cancer screening data. The updated results are reported in Fig. 3. We also increased the number of repeated runs from 5 to 10 for more reliable estimates. In addition, we added an adversarial random corruption experiment in Appendix D.3, where 30% of the original edges are randomly added or deleted. The results show that MaskGT improves robustness under structural perturbations, further supporting its ability to learn task-adaptive attention connectivity rather than relying solely on the given graph structure.

---

### Author Response · Authors · 2026-06-01

We thank the reviewers for their constructive feedback. We have uploaded a revised manuscript with the changes in green. The revision strengthens the empirical evaluation, robustness analysis, and ablation study. In particular, we have:

- Added experiments with additional Graph Transformer backbones in Sec. 5.2.
- Added ablation studies on key hyperparameters in Sec. 5.5.
- Increased the number of repeated runs and added NCI1 as an additional graph-classification dataset in Fig. 3.
- Added p-value analyses in Appendix D.2 to assess statistical significance of Fig. 3 experiments.
- Added adversarial random graph corruption experiments in Appendix D.3.

We hope these revisions address the reviewers’ concerns and provide stronger support for the claims of the paper.

---

### Decision · Action_Editor_7cUW · 2026-07-06

**Recommendation:** Accept as is

**Audience:**

Yes

**Audience Explanation:**

This work addresses an active research topic in graph machine learning by proposing a lightweight, modular approach for learning task-adaptive connectivity in Graph Transformers. The paper is relevant to researchers interested in Graph Transformers, graph representation learning, attention mechanisms, structural inductive biases, and parameter-efficient adaptation. Although the contribution is incremental rather than transformative, I believe it will be of interest to a meaningful portion of the TMLR readership.

**Claims And Evidence:**

Yes

**Claims Explanation:**

The revised manuscript provides sufficient evidence to support its main claims. During the review process, reviewers raised concerns regarding the evidence for the GT-agnostic claim, the statistical reliability of the reported improvements, the experimental validation, and the motivation for explicit masking. The authors addressed these concerns through a substantial revision, including additional experiments on multiple Graph Transformer backbones, expanded ablation studies, increased numbers of repeated runs, statistical significance analyses, and additional robustness experiments. These revisions substantially strengthen the empirical support for the proposed method.

While some reviewers continue to question the magnitude of the empirical gains and would have appreciated stronger analyses comparing explicit masking with learned attention patterns, these remaining concerns primarily relate to the strength and novelty of the contribution rather than the soundness of the presented evidence. Overall, I believe the claims made in the revised submission are supported by convincing and sufficiently clear empirical evidence.